# Cumulative Burden of Being Underweight Increases the Risk of Hip Fracture: A Nationwide Population-Based Cohort Study

**DOI:** 10.3390/healthcare10122568

**Published:** 2022-12-17

**Authors:** Han-Dong Lee, Sangsoo Han, Hae-Dong Jang, Kyungdo Han, Nam-Su Chung, Hee-Woong Chung, Ki-Hoon Park, Ha-Seung Yoon, Jae-Young Hong

**Affiliations:** 1Department of Orthopaedic Surgery, Ajou University School of Medicine, Suwon 16499, Republic of Korea; 2Department of Emergency Medicine, Soonchunhyang University Bucheon Hospital, Bucheon 14584, Republic of Korea; 3Department of Orthopaedic Surgery, Soonchunhyang University Bucheon Hospital, Bucheon 14584, Republic of Korea; 4Department of Statistics and Actuarial Science, Soongsil University, Seoul 06978, Republic of Korea; 5Department of Orthopedics, Korea University Hospital, Ansan 15355, Republic of Korea

**Keywords:** hip fracture, underweight, smoking, sex

## Abstract

(1) Background: Being underweight is a known risk factor for hip fractures. However, it is unclear whether the cumulative underweight burden affects the incidence of hip fractures. Therefore, we explored the effect of the cumulative underweight burden on the development of hip fractures; (2) Methods: In a cohort of adults aged 40 years and older, 561,779 participants who were not underweight and had no hip fractures from 2007 to 2009 were identified. The risk of hip fracture from the time of the last examination to December 2018 according to the cumulative burden of being underweight (based on 0 to 3 examinations) was prospectively analyzed; (3) Results: During follow-up (mean 8.3 ± 0.8 years), the prevalence of newly diagnosed hip fractures was 0.2%, 0.4%, 0.5%, and 0.9% among those with 0, 1, 2, and 3 cumulative underweight, respectively. The adjusted hazard ratios (HRs) with 95% confidence intervals (CIs) of groups meeting the diagnostic criteria for underweight 1, 2, and 3 compared to 0 were 2.3 (1.6–3.3), 2.9 (1.8–4.5), and 4.5 (3.4–6.1), respectively (*p* for trend < 0.01); (4) Conclusions: The risk of hip fracture increased as the burden of underweight accumulated.

## 1. Introduction

Hip fractures have become a significant health problem [1], and are particularly significant contributors to morbidity and mortality [2]. More than half of patients with hip fractures are unable to return to their pre-fracture condition [3]. The risk of death in the 3 months following a hip fracture increases by fivefold for women and eightfold for men relative to age- and sex-matched controls [4]. Thus, the socioeconomic burden imposed by hip fractures is very high. Johnell estimated that, in 1990, the cost of hip fractures worldwide was 34,900 million dollars (34.9 US billion dollars), with an approximate cost per patient of 21,000 dollars in the first year, and the projected worldwide costs 131,500 million dollars (131.5 US billion dollars) in 2050 [5]. Therefore, a substantial effort has been devoted to identifying clinical risk factors.

Current risk assessments are principally based on bone mineral density (BMD) measured via dual-energy X-ray absorptiometry (DXA) [6,7]. Apart from BMD, other “modifiable” factors have been analyzed; the body mass index (BMI) and body weight are key risk factors. One large-scale observational study found that the risk of hip fracture increased as BMI decreased because subjects with high BMI often exhibit a high BMD, strong bone structure, and a protective effect of fat [8,9]. The cited studies focused on the association between being underweight at one timepoint and the later occurrence of hip fracture. Both a single record of being underweight and accumulated records may increase the risk for disease [10]. The authors of this study studied the relationship between being underweight and hip fractures using a large-scale cohort, and reported the degree of underweight at one time or the risk of hip fractures according to changes in underweight [11,12,13]. However, the relationship between the accumulation of the underweight burden and hip fractures has not been studied. Therefore, we explored the effect of the cumulative underweight burden on the development of hip fractures.

## 2. Materials and Methods

### 2.1. Study Population

This study was a nationwide retrospective cohort study. We defined a population-based cohort using both the Korean National Health Insurance Service (K-NHIS) and the Korean National Health Examination (K-NHE) claims databases that cover the entire population of the Republic of Korea. All insured Koreans aged 40 years and older and all workers aged 20 years and older must undergo regular (free) K-NHE checkups every 1–2 years. The data collected include anthropometric measurements, smoking and alcohol consumption status, and medical histories (using both self-reported questionnaires and laboratory findings). The K-NHIS claims database includes diagnosis and comorbidities (coded using the International Classification of Diseases [ICD-10] 10th Edition), demographic data, prescriptions, medical services (treatments and procedures), inpatients and outpatient costs from all Korean clinics and hospitals.

We initially enrolled 629,738 individuals aged ≥40 years who underwent three consecutive annual K-NHIS health screenings between January 2007 and December 2009. We excluded individuals with missing data or who had a fracture within 1 year from the baseline examination in the index year. A total of 561,779 participants were ultimately included (Figure 1).

### 2.2. Evaluation of Being Underweight and the Cumulative Burden of Being Underweight

At each K-NHE examination, weight and height were measured and the BMI was calculated as weight/(height)^2. At the index health examination, we classified all subjects by BMI status: <18.5 (underweight), <23 (healthy), <25 (near overweight), <30 (overweight), and ≥30 (obese) using the recognized Asian criteria [14,15,16]. The total study population was divided into groups for whom underweight was diagnosed at 0 to 3 examinations.

### 2.3. Hip Fracture Assessment

K-NHIS medical claims records were used to identify hip fractures from the last examination to 31 December 2018 (Figure 2); we examined hospitalization records and the ICD-10 codes. Hip fractures were identified with the S72.0, S72.1, or S72.2 codes. Participants who died during the follow-up period were censored at the time of death.

### 2.4. Covariates

The baseline characteristics were those of the last health examination (on the index date) and included age (<65 and ≥65 years), sex, smoking status (nonsmoker, ex-smoker, and current smoker), alcohol consumption (non-drinker, mild-to-moderate drinker, and heavy drinker), income status (low vs. control), regular exercise status, and comorbidities (diabetes [DM], hypertension [HTN], dyslipidemia [DYS], and chronic kidney disease [CKD]). Alcohol consumption was defined as: non-drinker, 0 g/day; mild-to-moderate drinker, >0 to 30 g/day; and heavy drinker, ≥30 g per day. Regular exercise was defined as >30 min of moderate-intensity exercise (e.g., brisk walking, tennis doubles, or leisurely bicycling) ≥5 times a week or >20 min of vigorous exercise (e.g., running, climbing, fast cycling, or aerobics) ≥ 3 times a week. Low income was defined as income below the 20th percentile. Individuals with DM were defined as either those prescribed antidiabetic drugs (ICD-10 codes E11–E14) or subjects with fasting blood glucose levels > 126 mg/dL. HTN was defined as a systolic/diastolic blood pressure greater than or equal to 140/90 mmHg or at least one claim annually for an antihypertensive agent (ICD-10 codes I10–I13, I15). DYS was defined as a total cholesterol level greater than or equal to 240 mg/dL or at least one claim annually for an antihyperlipidemic agent (ICD-10 code E78). CKD was defined as an estimated glomerular filtration rate < 60 mL/min/1.73 m^2^. Blood pressure was measured using a blood pressure monitor after at least 5 min of res. All blood tests were performed after 8 h of overnight fasting.

### 2.5. Statistical Analysis

Data are reported as means ± standard deviations for continuous variables and numbers (%) for categorical variables. The incidence rate (IR) was the outcome per 1000 person-years (PY). The total number of hip fractures was divided by the total follow-up period of all individuals. We used the chi-square test to compare categorical variables and Student’s t-test to compare continuous variables We calculated the hazard ratios (HRs) with 95% confidence intervals (CIs) for hip fractures by the BMI at the time of the index health examination using Cox’s regression analysis. We constructed 4 models to explore covariates potentially associated with hip fractures. Model 1 was unadjusted. Model 2 was adjusted for sex and age. Model 3 was additionally adjusted for current smoking, heavy drinking, a low income, and regular exercise. Model 4 was additionally adjusted for DM, HTN, DYS, and CKD. We calculated HRs with 95% CIs for hip fractures by underweight cumulative status using Cox’s regression analysis. To investigate the effect of clinical condition on the association between the accumulation of underweight status and the risk of hip fracture, the HRs for hip fractures in the different subgroups were derived from Cox’s regression analysis, as were the *p*-values for the interaction. We performed stratified subgroup analysis by sex, age, current smoker, and heavy drinker status, low income, regular exercise, and the presence of comorbidities (DM, HTN, DYS, and CKD). All statistical analyses were performed using SAS software (ver. 9.3; SAS Institute, Cary, NC, USA). A two-tailed *p*-value < 0.05 was considered to indicate statistical significance.

## 3. Results

### 3.1. Baseline Characteristics (Table 1)

The baseline characteristics of 561,779 participants grouped by the number of underweight scores at the three health examinations are summarized in Table 1. Individuals were divided into four groups: 545,824 (97.2%), 6929 (1.2%), 3672 (0.7%), and 5354 (1.0%) who met the diagnostic criteria of being considered underweight 0, 1, 2, and 3 times, respectively. Age, sex ratio, smoking and alcohol consumption, regular exercise status, income, and comorbidities (DM, HTN, DYS, and CKD) differed significantly among the groups (all *p* < 0.01) because the population was large.

**Table 1 healthcare-10-02568-t001:** Baseline characteristics of participants according to the burden of underweight.

Underweight Burden	0 Time *n* = 545,824	1 Time *n* = 6929	2 Times *n* = 3672	3 Times *n* = 5354	*p*-Value
**Age**	49.7 ± 7.1	49.7 ± 7.8	49.7 ± 7.8	49.6 ± 7.9	0.68
<65	523,027 (95.8)	6529 (94.2)	3447 (93.9)	5044 (94.2)	**<0.01**
65≤	22,797 (4.2)	400 (5.8)	225 (6.1)	310 (5.8)	
**Sex**					**<0.01**
Male	403,926 (74)	4270 (61.6)	2373 (64.6)	3495 (65.3)	
Female	141,898 (26)	2659 (38.4)	1299 (35.4)	1859 (34.7)	
**Smoke**					**<0.01**
Non	257,094 (47.1)	3699 (53.4)	1808 (49.2)	2683 (50.1)	
Ex	127,920 (23.4)	960 (13.9)	515 (14)	580 (10.8)	
Current	160,810 (29.5)	2270 (32.8)	1349 (36.7)	2091 (39.1)	
**Drink ^a^**					**<0.01**
Non	234,442 (43)	3738 (54)	1916 (52.2)	2817 (52.6)	
Mild	264,341 (48.4)	2826 (40.8)	1539 (41.9)	2247 (42)	
Heavy	47,041 (8.6)	365 (5.3)	217 (5.9)	290 (5.4)	
**Regular exercise ^b^**	122,611 (22.5)	1010 (14.6)	476 (13)	655 (12.2)	**<0.01**
**Low-Income ^c^**	112,089 (20.5)	1570 (22.7)	823 (22.4)	1222 (22.8)	**<0.01**
**Comorbidity**					
Diabetes	51,701 (9.5)	409 (5.9)	205 (5.6)	244 (4.6)	**<0.01**
Hypertension	157,011 (28.8)	1052 (15.2)	490 (13.3)	641 (12)	**<0.01**
Dyslipidemia	103,324 (18.9)	620 (9)	291 (7.9)	352 (6.6)	**<0.01**
Chronic kidney disease	42,114 (7.7)	378 (5.5)	202 (5.5)	315 (5.9)	**<0.01**
**Hip fracture**	995 (0.2)	30 (0.4)	20 (0.5)	46 (0.9)	**<0.01**

Means ± standard deviation. Categorical variables: *n* (%). Bold indicates a statistically significant result (*p* < 0.05). ^a^ Alcohol consumption denotes as following; Non-drinker, alcohol consumption 0 g; Mild to moderate drinker, alcohol consumption > 0 g to 30 g per day; Heavy drinker: alcohol consumption ≥ 30 g per day. ^b^ Regular exercise denotes performing > 30 min of moderate-intensity exercise (e.g., brisk pace walking, tennis doubles, or bicycling leisurely) ≥ 5 times a week or >20 min of vigorous-intensity exercise (e.g., running, climbing, fast cycling, or aerobics) ≥ 3 times a week. ^c^ Low income denotes income belongs to lower 20% among entire Korean population of subjects supported by the Medical Aid program.

### 3.2. Risk of Hip Fracture According to BMI at the Index Health Examination (Table 2)

At the index health examination, the population was divided into five groups according to BMI: 10,121 (1.8%), 201,152 (35.8%), 159,052 (28.3%), 177,198 (31.5%), and 14,256 (2.5%) who were underweight, healthy, near overweight, overweight, and obese, respectively. During follow-up (mean 8.3 ± 0.8 years), the IRs per 1000 PY of newly diagnosed hip fracture were 0.9, 0.3, 0.2, 0.2, and 0.3, respectively. Adjusted Cox’s proportional hazards regression analyses were performed to calculate adjusted HRs (model 4) for newly diagnosed hip fractures by the BMI at the index health examination. Underweight was associated with a significantly higher risk despite the adjustment for several potentially confounding variables (underweight group: adjusted HR = 2.8, 95% CI = 2.2–3.6; *p* < 0.01).

**Table 2 healthcare-10-02568-t002:** The risk of hip fracture according to body mass index.

BMI	Event (*n*), Fracture	Total FU Duration (PY)	IR (per 1000 PY)	Hazard Ratio (95% CI)			
				Model 1	Model 2	Model 3	Model 4
**<18.5 (underweight)** ***n* = 10,121, 1.8%**	72	82,370.67	0.9	3.1 (2.4,4.0)	2.8 (2.2,3.6)	2.7 (2.1,3.4)	2.8 (2.2,3.6)
**<23 (healthy)** ***n* = 201,152, 35.8%**	468	1,664,441.74	0.3	1 (Ref.)	1 (Ref.)	1 (Ref.)	1 (Ref.)
**<25 (risk-to-overweight)** ***n* = 159,052, 28.3%**	249	1,318,498.97	0.2	0.7 (0.6,0.8)	0.6 (0.6,0.8)	0.7 (0.6,0.8)	0.6 (0.5,0.7)
**<30 (overweight)** ***n* = 177,198, 31.5%**	265	1,467,807.3	0.2	0.6 (0.6,0.7)	0.6 (0.5,0.7)	0.6 (0.5,0.7)	0.6 (0.5,0.7)
**30≤ (obese)** ***n* = 14,256, 2.5%**	37	117,544.35	0.3	1.1 (0.8,1.6)	1.2 (0.9,1.7)	1.2 (0.9,1.7)	1.0 (0.7,1.4)
***p* value**				<0.01	<0.01	<0.01	<0.01

Abbreviations: FU follow-up; PY person-year; IR incidence rate; CI confidence interval. Incidence rate = event (fracture)/total follow-up duration. Model 1 was unadjusted. Model 2 was adjusted for age and sex. Model 3 was additionally adjusted for smoking status, alcohol consumption, income status, and regular exercise. Model 4 was additionally adjusted for diabetes status, hypertension, dyslipidemia, and chronic kidney disease.

### 3.3. Accumulation of Underweight Burden and the Risk of Hip Fractures (Table 3)

During the follow-up, the IRs per 1000 PY of newly diagnosed hip fractures were 0.2, 0.5, 0.7, and 1.1 in those who met the underweight diagnostic criteria 0, 1, 2, and 3 times, respectively. The incidence of hip fractures increased as the cumulative number of underweight records increased (*p* < 0.01). The incidences of hip fracture by the number of underweight records are listed in Table 3. The risk of fracture was positively associated with the cumulative number of underweight records: the adjusted HRs (model 4) with 95% CIs of groups meeting the diagnostic criteria of being underweight 1, 2, and 3 times (compared to never) were 2.3 (1.6–3.3), 2.9 (1.8–4.5), and 4.5 (3.4–6.1), respectively (*p* for trend < 0.01).

**Table 3 healthcare-10-02568-t003:** The risk of hip fracture according to underweight burden.

Underweight Burden	Event (*n*), Fracture	Total FU Duration (PY)	IR (per 1000 PY)	Hazard Ratio (95% CI)			
				Model 1	Model 2	Model 3	Model 4
**0 ** **(*n* = 545,824, 97.16%)**	995	4,520,214.53	0.2	1 (Ref.)	1 (Ref.)	1 (Ref.)	1 (Ref.)
**1** **(*n* = 6929, 1.23%)**	30	56,854.27	0.5	2.4 (1.7, 3.5)	2.2 (1.5, 3.2)	2.1 (1.5, 3.1)	2.3 (1.6, 3.3)
**2** **(*n* = 3672, 0.66%)**	20	30,009.1	0.7	3.0 (2.0, 4.7)	2.8 (1.8, 4.4)	2.7 (1.7, 4.2)	2.9 (1.8, 4.5)
**3** **(*n* = 5354, 0.95%)**	46	43,585.13	1.1	4.8 (3.6, 6.5)	4.5 (3.3, 6.0)	4.2 (3.1, 5.6)	4.5 (3.4, 6.1)
***p* value**				<0.01	<0.01	<0.01	<0.01

Abbreviations: FU follow-up; PY person-year; IR incidence rate; CI confidence interval. Incidence rate = event (fracture)/total follow-up duration. Model 1 was unadjusted. Model 2 was adjusted for age and sex. Model 3 was additionally adjusted for smoking status, alcohol consumption, income status, and regular exercise. Model 4 was additionally adjusted for diabetes status, hypertension, dyslipidemia, and chronic kidney disease.

### 3.4. Subgroup Analyses of the Risk of Fracture by Accumulation of Underweight Burden Records

Multivariable Cox’s proportional hazards regression analyses were adjusted for confounding variables (age, sex, current smoking, heavy drinking, regular exercise, low income, and comorbidities) and used to estimate adjusted HRs for hip fracture. The incidence of hip fracture did not exhibit any interaction with age, heavy drinking, regular exercise, low income, DM, HTN, DYS, or CKD (all *p* for interactions > 0.05). The effect of being underweight on hip fracture occurrence was significantly higher in men than women; the HRs of those with 1, 2, and 3 underweight records (men vs. women) were 2.6 vs. 1.7, 3.7 vs. 1.1, and 5.8 vs. 1.8, respectively. The effect of the underweight burden on hip fracture occurrence was significantly higher among current smokers than non-smokers; the HRs of those with 1, 2, and 3 underweight records (current smokers and non-smokers) were 2.9 vs. 1.9, 4.3 vs. 2.0, and 6.2 vs. 3.3, respectively.

## 4. Discussion

To the best of our knowledge, this is the first study to report the cumulative burden of being underweight on the risk of hip fracture in a large national population. Our principal finding was that the cumulative burden of being underweight over the three health examinations correlated linearly with the risk of hip fracture. Even after adjusting for age, sex, smoking status, alcohol consumption, income status, regular exercise, and comorbidities (DM, HTN, DYS, and CKD), the results were consistent. The relationships were strongest in males and current smokers.

Being underweight is a recognized and preventable risk factor for hip fractures [17,18]. A lower BMI was associated with a lower BMD [19]. Lloyd et al. found that every unit increase in BMI is associated with a BMD increase of 0.0082 g/cm [20]. However, being underweight increases the risk of hip fracture regardless of BMD [8]. As the BMI increases, adipose tissue increases, affording to cushion against fracture [21]. We found that hip fracture incidence was about 2.8-fold higher in the group that was underweight at baseline than the normal group. Recent Asian studies have reported that the risks of hip fracture in underweight groups are about 1.7- to 3.9-fold higher than in normal groups; our figures are similar [18,22,23].

We also found that the risk of hip fractures increased as the cumulative period of being underweight increased. When being underweight was recorded 1, 2, and 3 times, the risk of hip fractures increased 2.3-, 2.9-, and 4.5-fold, respectively. We are the first to report that the accumulation of the underweight burden increases the risk of hip fractures. This constitutes strong evidence that being underweight must be corrected to prevent hip fractures. Park et al. reported that not only being underweight, but also the accumulation of the underweight burden, weakened the immune system and increased the risk for tuberculosis [10]. Although no report has indicated that underweight accumulation accelerates a decrease in bone density, the literature indicates that being underweight reflects undernutrition; it is likely that bone density will worsen if undernutrition persists, increasing the risk of fracture [24]. In addition, prolonged exposure to trauma in the absence of a fat cushion may affect the risk of fracture.

Some studies have found that the impact of BMI on fractures varied by sex [8,23,25]. In a recent large-scale study in Japan, being underweight was a risk factor for hip fracture regardless of sex, and was a risk factor for vertebral fracture only in men [23]. Our results are in line with these findings; regardless of sex, being underweight was a risk factor for hip fracture. Subgroup analysis revealed that the risk of hip fracture attributable to being underweight was significantly higher in men than women, perhaps because the fat distribution varies by sex; women are better cushioned [26].

We confirmed that smoking increases the risk of hip fracture. In the relevant sub-analysis, the incidence of hip fracture was 1.5- to 2.3-fold higher in current smokers than non-smokers. Recent studies that have used large databases from several countries have confirmed that smoking is a risk factor for hip fracture [27,28]. In some previous studies, the hip fracture rate was 1.3- to 3.3-fold higher in current smokers than non-smokers; our results agree with this [27,28,29]. Smoking may compromise BMD and be associated with a leaner body mass and a less active lifestyle [30,31].

This study had several strengths; the principal one was the inclusion of various age groups, including both adults and the elderly. In addition, fracture incidence rates were from the national medical claims records, which were more accurate than retrospective data. We also adjusted for potential confounding factors (age, gender, current smoking, alcohol consumption, low income, and comorbidities). We also performed extensive subgroup analyses. The limitations include the retrospective design (we used data from a claims database and medical examinations that lacked BMD values). However, BMI affects the hip fracture incidence independent of the BMD, and the BMI can be very simply determined. Second, the proportion of underweight patients was very small (about 1%) because all patients who had previously been underweight were excluded [25]. Third, some inaccuracies are inevitable with large cohort databases. An accurate operating definition of the code for hip fracture is important. In this study, the operating definition of several previously published studies was referred to [11,12,13,32,33]. Finally, caution is warranted when generalizing these results to countries other than Korea. Future multiethnic cohort studies are needed.

## 5. Conclusions

In this study, through a large-scale cohort study, it was proved that being underweight increases the risk of hip fractures. In addition to that, this study discovered that the risk of hip fracture increased according to the accumulation of underweight. If a subject at risk for hip fracture is underweight, it is important to correct this quickly, particularly in males and current smokers.

## Figures and Tables

**Figure 1 healthcare-10-02568-f001:**
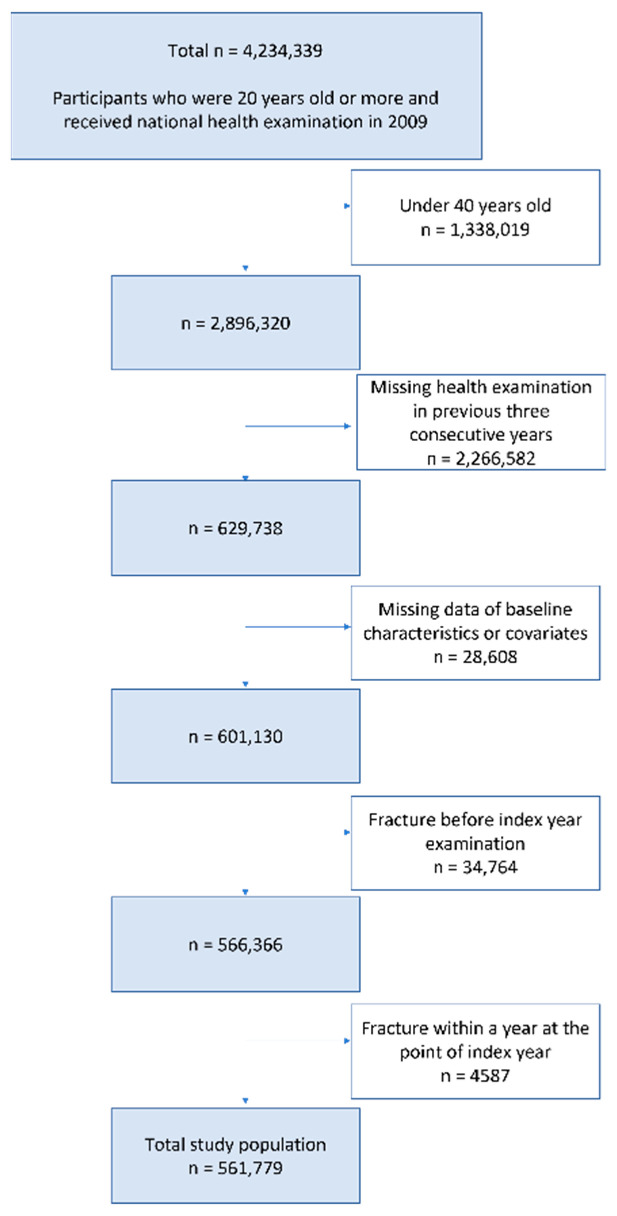
Flowchart of the inclusion and exclusion criteria.

**Figure 2 healthcare-10-02568-f002:**
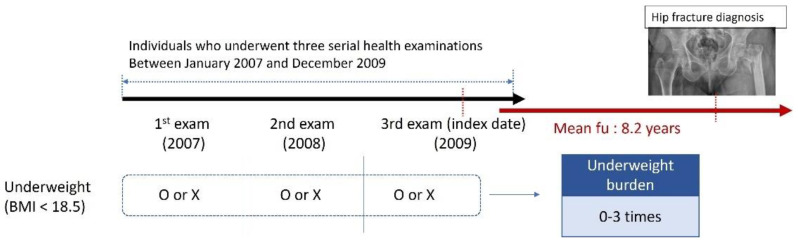
Overall plan of the study. Evaluation of the cumulative underweight burden (0–3 records) at three consecutive health examinations (between 2007 and 2009) by incidence of hip fracture during a mean follow-up of 8.2 years.

## Data Availability

Data available from the authors of this publication.

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
