# Peer review of "Cumulative Burden of Being Underweight Increases the Risk of Hip Fracture: A Nationwide Population-Based Cohort Study"

_healthcare, 2022, doi:10.3390/healthcare10122568_

Round 1
Reviewer 1 Report
Manuscript number: healthcare-2083582
Review of the manuscript titled: Cumulative burden of being underweight increases the risk of hip fracture: A nationwide population-based cohort study
Article type: Prospective cohort study
Comments
In this study entitled “Cumulative burden of being underweight increases the risk of hip fracture: A nationwide population-based cohort study”, the authors investigated cumulative burden of being underweight related hip fracture. They concluded that the risk of hip fractures is increased with accumulation of underweight burden. This manuscript is easily read and the message is straightforward. However, there are some concerns about the backgroud and study design.
- The methodology and interpretation of the results of various studies submitted by your research team to other journals are nothing new compared to this study. Therefore, please mention and cite similar papers published by your team.
- It is well known that low BMI is an important factor in increasing fracture risk. Please describe in more detail how this paper differs from previous papers.
- This study is a study using only code. Is the operational definition accurate in the study using it?
- Has the operating definition of the study been validated?
Author Response
We appreciate your kind review of our manuscript.
- Our team published three papers on the relationship between hip fractures and being underweight. The difference between this study and previous studies is that previous studies focused on the degree of underweight at a point in time, while this study focused on the accumulation of underweight. We have added this in the Introduction section.
- In this study, previous studies that showed a correlation between BMI and hip fracture were re-proven through a large-scale study. In addition to that, it was newly proved that the accumulation of underweight increases the risk of hip fracture. The conclusion of the paper has been modified to emphasize this point more clearly.
- The authors acknowledge that operating definition is important for a study using only codes such as this study. This study used the operating definition used in other published studies. This is in addition to the limitations in the discussion section.
Reviewer 2 Report
This is not the most relevant topic in the field from the orthopaedic surgeon s prospective like myself, but it is an important one because: -it is a nation wide study in the country where the hip fractures incidence is constantly raising, and that has an extremely high elderly population growth rate and prolonged life expectance, -the country in which, according to previously published data, the age adjusted prevalence of underweight among adults is from 3.1% in males to 6,3% in women ( which is more or less the same like in USA and EU) Considering the fact that there is a constant trend in raising numbers of underweight adults above 50 years of age worldwide, the results of this study could give a significant new informations on this matter to the readers.
This is the study that analyzed the cumulative burden of being underweight and the risk of hip fractures based on the well known fact that underweight is a risk factor for increased mortality, higher fracture incidence, osteoporosis etc. 4. The authors (or at list some of them) have recently published several articles about different aspects of the underweight and hip fractures relation. Seems to me that their methodology is well designed and appropriate.
Author Response
We appreciate your kind review of our manuscript.